

# Prediction of soil moisture using BiGRU-LSTM model with STL decomposition in Qinghai–Tibet Plateau

Lufei Zhao[1], Tonglin Luo[2], Xuchu Jiang[2] and Biao Zhang[3]

[1] Agricultural Science and Engineering School, Liaocheng University, Liaocheng, Shandong, China
[2] School of Statistics and Mathematics, Zhongnan University of Economics and Law, Wuhan, Hubei, China
[3] School of Computer Science, Liaocheng University, Liaocheng, Shandong, China

Corresponding author
Xuchu Jiang,
xuchujiang@zuel.edu.cn

## ABSTRACT

Ali Network data based on the Qinghai-Tibetan Plateau (QTP) can provide representative coverage of the climate and surface hydrometeorological conditions in the cold and arid region of the QTP. Among them, the plateau soil moisture can effectively quantify the uncertainty of coarse resolution satellite and soil moisture models. With the objective of constructing an "end-to-end" soil moisture prediction model for the Tibetan Plateau, a combined prediction model based on time series decomposition and a deep neural network is proposed in this article. The model first performs data preprocessing and seasonal-trend decomposition using loess (STL) to obtain the trend component, seasonal component and random residual component of the original time series in an additive way. Subsequently, the bidirectional gated recurrent unit (BiGRU) is used for the trend component, and the long short-term memory (LSTM) is used for the seasonal and residual components to extract the time series information. The experiments based on the measured data demonstrate that the use of STL decomposition and the combination model can effectively extract the information in soil moisture series using its concise and clear structure. The proposed model in this article has a stable performance improvement of 5–30% over a single model and existing prediction models in different prediction time domains. In long-range prediction, the proposed model also achieves the best accuracy in the shape and temporal domains described by using dynamic time warping (DTW) index and temporal distortion index (TDI). In addition, the generalization performance experiments show that the combined method proposed in this article has strong reference value for time series prediction of natural complex systems.

# INTRODUCTION

## Background

As the highest plateau in the world, the Qinghai-Tibet Plateau (QTP) is an important ecological security barrier for the world, playing many roles in water conservation and biodiversity protection. As an important indicator of surface hydrological information, soil

moisture plays an important role in regional energy and the land water cycle (*Milly P.C. & Dunne, 1994*) and is an important parameter in hydrological, meteorological and environmental studies. Its temporal variation and spatial distribution regulate the pattern, diversity and succession characteristics of vegetation (*Zhu et al., 2017*). The main grassland type on the QTP is alpine grassland, and the soil moisture in the root layer is mainly affected by rainfall recharge factors. Therefore, an in-depth understanding of soil water dynamics is helpful to better understand soil water maintenance and predict the potential impact of future rainfall pattern changes on key processes of alpine steppe ecosystems (*Xing et al., 2009*). It is of great significance to study the spatial and temporal variation pattern of surface soil moisture on the QTP and build a soil moisture prediction model based on long-term time series data for the study of alpine grassland ecological carrying capacity, ecological construction of grassland restoration and reconstruction, and meteorological disaster monitoring in the QTP.

## Literature review

Time series generated by complex systems are commonly found in various fields, such as astronomy, hydrology, meteorology, environment, and finance. These time series often exhibit highly intricate nonlinear characteristics and manifest as multivariate and large-scale in nature. At the same time, the data are characterized by nonstationarity and noise due to the complex evolution of the system and external disturbances (*Han et al., 2019*). Traditionally, the time series involved in these areas have been modelled and predicted using numerical models, and *Su et al. (2013)* developed a numerical prediction model for soil moisture content on the QTP using a series of interpolation methods and a time-point-by-time extended Kalman filter based on the basic framework given by the European Centre for Medium-Range Weather Forecasting (ECMWF), with significant performance improvements over existing numerical models. However, the generalization of the numerical methods is limited, necessitating the expenditure of considerable time designing intricate mathematical models to address different scenarios. It also imposes a significant computational burden.

In recent years, with the development of data science and measurement techniques, soil moisture prediction models that are entirely driven by data have become progressively more abundant. Data-driven models strive to approximate complex real-world situations as closely as possible by leveraging extensive data, and they have found wide applications in the domain of complex system time series. Some researchers have proposed soil moisture prediction models that integrate multiple sources of data. For instance, *Togneri et al. (2022)* introduced a model based on LightGBM and sensor network data, while *Luo, Wen & He (2023)* proposed a model based on back propagation (BP) neural networks and optical and thermal infrared (TIR) spectroscopy. *Zhu et al. (2023)* presented a model based on random forests (RF) and climate observation data such as evapotranspiration, and *Yin, Wang & Huang (2023)* proposed a method based on support vector machines (SVM) and soil state data such as soil temperature. Moreover, several researchers have explored the integration of various observation data from different sources, including satellite data, sensor data, and *in situ* data, to establish numerous soil moisture prediction models for

diverse application scenarios. These models utilize deep learning methods such as residual learning (*Li et al., 2022a*), long short term memory (LSTM) (*Filipović et al., 2022*), convolutional neural network (CNN) and bidirectional gated recurrent unit (BiGRU) (*Yuan et al., 2022*), the combination of attention mechanism and LSTM (*Li et al., 2022b*), as well as a specially designed artificial neural network (ANN) (*Singh & Gaurav, 2023*). Although these models have achieved high accuracy, they still require laborious feature engineering. Additionally, the availability of data severely limits the practical application of these models since they rely on large amounts of additional data as inputs.

Establishing an "end-to-end" soil moisture prediction model holds promise for effectively addressing the above issues. Traditional statistical learning methods for time series, also known as modern time series analysis, originated from the autoregressive (AR) model proposed by statisticians in 1927. In the 1970s, the autoregressive integrated moving average (ARIMA) model became the central topic of time series analysis. Some studies have proposed combining the ARIMA model with the BP neural network model to simultaneously consider the linear and nonlinear characteristics of soil moisture data, resulting in improved predictive performance compared to using a single model (*Wang, Han & Chang, 2023a*). Furthermore, *Wang et al. (2023b)* incorporated the GRU model into block Hankel tensor ARIMA, achieving even better results. However, ARIMA models and their various derivatives struggle to handle complex cyclic and trend changes in soil moisture prediction. Therefore, several time series decomposition methods have been used in the study of complex system time series, including singular value decomposition (SVD) (*Liu, 2003*), principal component analysis (PCA) (*Chitsaz, Azarnivand & Araghinejad, 2016*), and wavelet decomposition (*Yang et al., 2018*). These decomposition methods to some extent extract the information inherent in soil moisture series, but they rely on strong mathematical assumptions that are often difficult to meet in practical scenarios. Empirical modal decomposition (EMD) and its derivatives, such as ensemble empirical mode decomposition (EEMD) and complete ensemble empirical mode decomposition with adaptive noise (CEEMDAN), can extract even more complex information from the series. *Prasad et al. (2019)* combined EEMD and extreme learning machine (ELM) to propose a short-term soil moisture prediction model based on multivariate sequences. However, similar models such as EEMD are designed to process signal sequences, and the numerous intrinsic mode functions (IMFs) produced by the decomposition reduce the interpretability of the model, significantly increase computational costs, and often suffer from issues such as mode mixing or incomplete decomposition due to random factors (*Qin, Li & Li, 2019*).

As a statistical method, seasonal-trend decomposition using loess (STL) exhibits good adaptability to various types of time series data with different properties. Models based on STL decomposition have demonstrated excellent performance in numerous fields of complex system time series prediction. *Ding et al. (2023)* combined STL with the random forest to investigate the influence of meteorological factors and precursor emissions on ozone concentrations. *Xu et al. (2022)* developed a framework called SDIPBC, which utilized STL and LSTM models to address and optimize sequence boundaries in

streamflow prediction. *Qin, Li & Li (2019)* applied STL to passenger flow prediction. STL decomposition demonstrates tremendous potential in soil moisture prediction problems.

## Contributions

This article introduces STL decomposition to the field of soil moisture prediction for the first time and proposes an "end-to-end" framework for soil moisture prediction, which enables highly accurate prediction of soil moisture content with insufficient information and without extensive additional feature engineering. In addition, its main advantages are as follows:

(1) The model has a concise and clear structure with low computational cost. The study demonstrates that STL decomposition effectively extracts trend and periodic information from the soil moisture content series. The proposed model further fits the subseries obtained from STL decomposition using a recurrent neural network model. Each component of the model has clear practical meaning and a concise structure compared to existing decomposition-prediction models. In addition, with the special design, parallel computation of the components can be achieved, further reducing the inference time.

(2) The proposed model exhibits good stability and generalization performance. The model consistently shows excellent prediction stability across various real-world scenarios tested over long time scales. At the same time, the proposed model has the best performance in terms of the *S* index, which measures the stability of different prediction steps of the soil moisture prediction models proposed in this article. In addition to soil moisture prediction, the proposed model also shows excellent results in soil heat flux prediction. This study provides significant guidance for predicting time series in complex natural systems.

(3) The multistep prediction values of the proposed model achieve the best performance in both temporal and morphological aspects. Since the accuracy of soil moisture content series in the temporal and morphological domains is crucial for subsequent analysis, experiments conducted in this study show that the proposed model performs the best in terms of the dynamic time warping (DTW) index and temporal distortion index (TDI). Thus, the proposed model has high practical value.

# DATA SOURCES AND RESEARCH METHODS

## Data sources and data preprocessing

The experiment to choose the soil moisture measured data from the National Qinghai-Tibet Plateau Scientific Data Center included the observation data of soil temperature and humidity of the QTP. The observational data in this dataset consist of four *in-situ* reference networks at regional scales, namely, the Naqu, Maqu, Ali and Pari networks with different climatic and vegetation types. The Ali Network, which includes the Ali and Shiquanhe regions, is located on the China-India border, approximately 1–2 km from the small village of Rutol and approximately 8 km from the inland lake Pangang Tso Lake. All soil moisture stations are distributed between 32°30′–33°30′ N and 79°50′–80°03′ E, at an altitude of approximately 4,260 m. The Ali Network is located in the arid southwestern part of the QTP, with an average annual precipitation of 87 mm, mainly

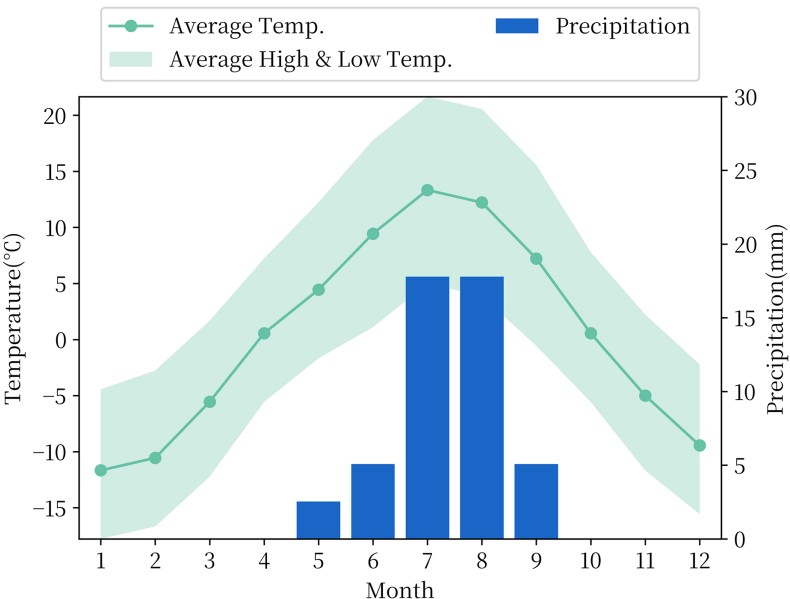

**Figure 1 Monthly temperature and precipitation in the Ali Network area.**

concentrated in summer, evaporation of 2,465 mm, an average annual temperature of 2 °C and mostly sunny days throughout the year, and its temperature and precipitation data by month are shown in Fig. 1. The landscape where the stations are located is mainly desert or sparse grassland. This type of landform covers approximately 23% of the total area of the QTP (*Fang et al., 2007*). At each station of the Ali Network, soil moisture content with an accuracy of $10^{-5}$ is recorded hourly at depths of 5, 10, 30, 50 and 80 cm. Based on previous research experience (*Yan & Wang, 2009*), it is known that microwave data can only reflect the surface soil moisture of a few centimetres, and considering that there is a large number of missing observational data of all sites of the Ali Network before 2011, in this article, soil moisture observation data recorded by the soil moisture sensor at a depth of 5 cm at the AL02 site of the Ali Network every 1 h between 2012 and 2016 were used for research.

This article divides the dataset according to the experience ratio of the training set and the test set of 8:2. Since the original data are time series data, the data are divided into the training set and the test set by taking 2015-9-16 0:00 as the partition node. Visualization of the training set and test set data is shown in Fig. 2. Finally, the sequence was normalized to map it to the interval (−1,1).

## Research methods

### STL decomposition

The STL decomposition proposed by *Cleveland et al. (1990)* decomposes the time series into trend, seasonal and remainder components. STL decomposition has good generality and robustness and is applicable to time series data of various cycles or frequencies. The core of the algorithm is to extract the seasonal trend information contained in the time

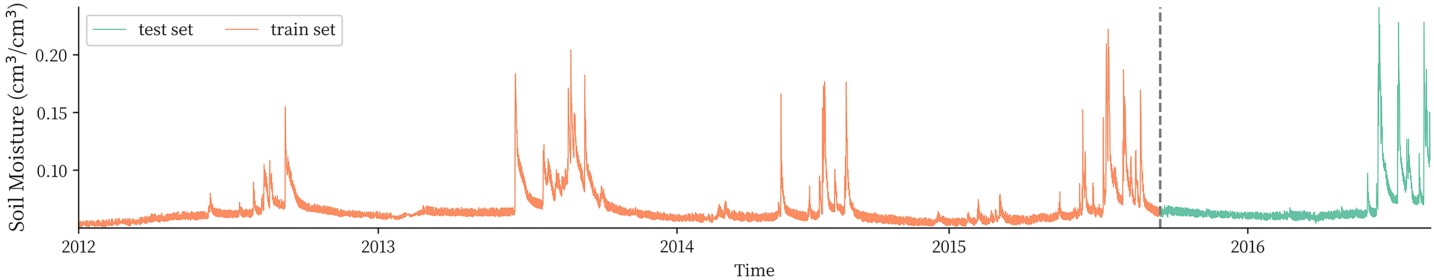

**Figure 2 Plateau soil moisture observation sequence and division of the training and test set.**

series more accurately by introducing local regression smoothing. STL decomposition represents the original sequence in the additive way as Eq. (1):

$$x_t = T_t + S_t + R_t \quad (t = 1, 2, 3, \ldots, N) \#() \tag{1}$$

where $T_t$ is the trend term, $S_t$ is the seasonal term, and $R_t$ is the remainder term.

The iterative process of the STL decomposition algorithm can be briefly described as follows:

(1) Set the initial iteration value: $k = 0$, $T_t^k = 0$.

(2) Detrending: $x_t - T_t^k$.

(3) Carry out smoothing on each detrended periodic subsequence, and the sequence obtained by combining all periodic subsequences is denoted as $C_t^{k+1}$.

(4) For $C_t^{k+1}$, low-pass filtering is carried out using the three times sliding average and once LOESS smoothing, $L_t^{k+1}$ is obtained.

(5) Calculate the seasonal terms: $Ss_t^{k+1} = C_t^{k+1} - L_t^{k+1}$.

(6) Calculate the trend term: The trend term $T_t^{k+1}$ is obtained by LOESS smoothing $x_t^k - S_t^{k+1}$.

(7) If $T_t^{k+1}$ converges or reaches the maximum number of iterations, the iteration terminates; otherwise, go back to step (2).

The decomposition process of STL is mainly controlled by parameters $n_p$, $n_s$ and $n_t$. The parameter $n_p$ is the cycle length in the sequence, and the smoothing parameter of the periodic subsequence $n_s$ is the parameter of the process in the third step. Generally, an odd number that is slightly larger than the number of cycles contained in the original sequence is taken. The trend smoothing parameter $n_t$ is the parameter of the LOESS process in the sixth step. Cleveland R B suggests a minimum odd number greater than $\dfrac{1.5 n_p}{1 - 1.5/n_s}$ (*Cleveland et al., 1990*).

### LSTM

The LSTM model is a kind of RNN model that was first proposed by *Hochreiter & Schmidhuber (1997)*, which can solve the gradient disappearance and gradient explosion problems faced by RNNs in the process of long time series (*Rakthanmanon et al., 2012*) and is specifically designed to avoid the long-term dependence problem (Fig. 3). Compared with the traditional RNN model, the LSTM model can perform better in a longer time series. The hidden layer of the original RNN has only one state, so it is very
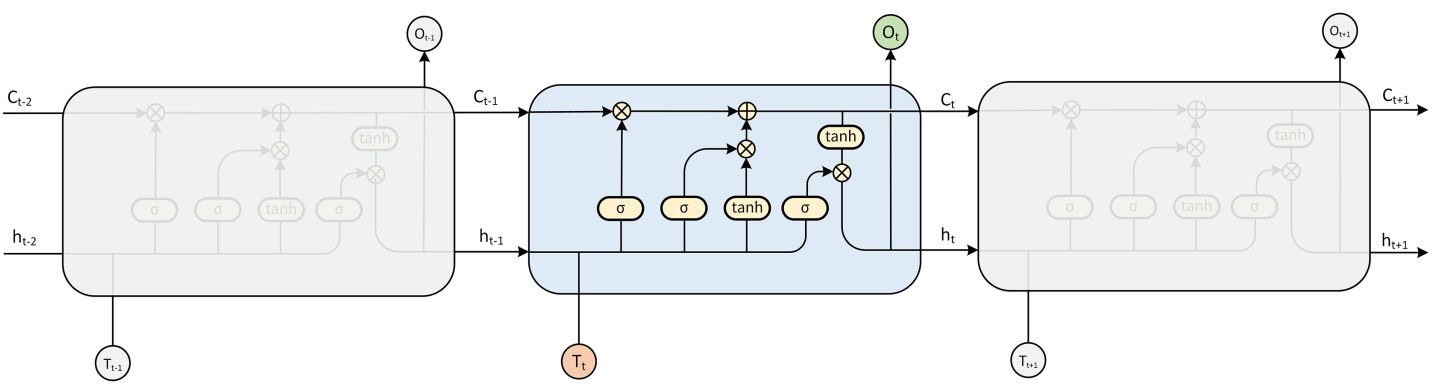

**Figure 3 LSTM model expansion diagram.**

sensitive to short-term input. The LSTM model adds another state based on the RNN, which is used to store the long-term state, called the cell state.

At the present moment, LSTM has three inputs: the current input value $x_t$, the output value of the LSTM at the previous moment $h_{t-1}$ and the cell state of the LSTM at the previous moment $C_{t-1}$. There are two outputs: the LSTM output value at the current moment $h_t$ and the cell state at the current moment $C_t$.

LSTM implements this mode through three gating mechanisms in the algorithm, namely, the input gate, forget gate and output gate. The input gate and output gate are used to receive, output, and correct parameters. The input gate determines how much of the network's input $x_t$ is saved to the cell state at the current time. The output gate determines how much of the cell state $C_t$ is output to the current output value $h_t$ of the LSTM. The forget gate determines how much of the cell state of the previous moment $C_{t-1}$ is retained to the cell state of the current moment $C_t$.

The LSTM determines the final output value $h_t$ as Eqs. (2)–(5). First, it calculates the activation state value $f_t$ of the forget gate at the current moment $t$:

$$f_t = \sigma\big(W_f \otimes (X_t h_{t-1}) + b_i\big)\#()$$ (2)

where $\sigma(\cdot)$ is the sigmoid function and $\otimes$ represents dot multiplication. After the vector is multiplied by the weight matrix, it is transformed by the activation function as a gated state.

Then, calculate the value of the input gate $i_t$ and the value of the candidate state of the input cell $\tilde{C}_t$ at moment $t$:

$$i_t = \sigma(W_i \otimes (X_t h_{t-1}) + b_i)$$ (3)

$$\tilde{C}_t = \sigma(W_i \otimes (X_t h_{t-1}) + b_i \#()$$ (4)

The updated value $\tilde{C}_t$ of the cell state under the current time $t$ can be obtained from the above calculation:

$$C_t = f_t \otimes C_{t-1} + i_t \otimes \tilde{C}_t \#()$$ (5)

Finally, calculate the current output value of the output gate according to the update value of the cell state at the current time $t$:

$$O_t = \sigma(W_0 \otimes (X_t h_{t-1}) + b_0) \tag{6}$$

$$h_t = O_t \otimes \tanh(C_t) \#() \tag{7}$$

### BiGRU

GRU is a simplification of the LSTM model proposed by *Cho et al. (2014)*. The LSTM model effectively alleviates the problem of gradient disappearance in the traditional RNN model. However, the shortcomings of the LSTM model, such as complex parameters and difficult training, are gradually exposed, restricting the further application of LSTM. The GRU redesigns the internal structure of the LSTM unit based on the gating idea, thus reducing the computation time and training complexity.

Similar to the LSTM model, for the input sequence $\{x_1, x_2, x_3, \ldots, x_t, \ldots x_n\}$, the GRU can successively obtain its hidden layer state $h_t$ at time step $t$ according to Eqs. (6)–(9):

$$r_t = \sigma(W_r x_t + b_r + W_{hr} h_{t-1} + b_{hr}) \#() \tag{8}$$

$$z_t = \sigma(W_z x_t + b_z + W_{hz} h_{t-1} + b_{hz}) \#() \tag{9}$$

$$n_t = \tanh(W_n x_t + b_n + r_t \otimes (W_{hn} h_{t-1} + b_{hn})) \#() \tag{10}$$

$$h_t = (1 - z_t) \otimes n_t + z_t \otimes h_{t-1} \#() \tag{11}$$

where $h_{t-1}$ is the hidden layer state of time step $t - 1$, $r_t, z_t, n_t$ is the gated state updated at each time step, $\sigma(\cdot)$ is a sigmoid function, and $b$ is the bias term.

BiGRU (bidirectional GRU) builds two reverse GRU models at the same time, modelling time sequence information forward and backwards, and the output of each time step is the concatenation of the output of the two GRU models. It is generally believed that the BiGRU model can better extract the front and back dependencies in time series and has a better effect for sequences with a certain front or back correlation (*Zhu et al., 2019*).

### Combined prediction model

Figure 1 shows that the observed data of soil moisture have a very significant seasonal variation rule with a 1-year cycle. Soil moisture in summer is much higher than that in the other three quarters, and the peak value of soil moisture in summer has a trend of gradual increase with the passage of time. Based on the nature of plateau soil moisture time series data, this article combined STL decomposition with the BiGRU model and LSTM model and proposed a new neural network combination prediction model based on STL decomposition to make use of the information extraction ability of STL decomposition and the time series fitting ability of the neural network model simultaneously. The overall framework of the model is shown in Fig. 4.

Based on a series of data preprocessing steps, the model first extracts the trend change information and periodic change information contained in the data through STL
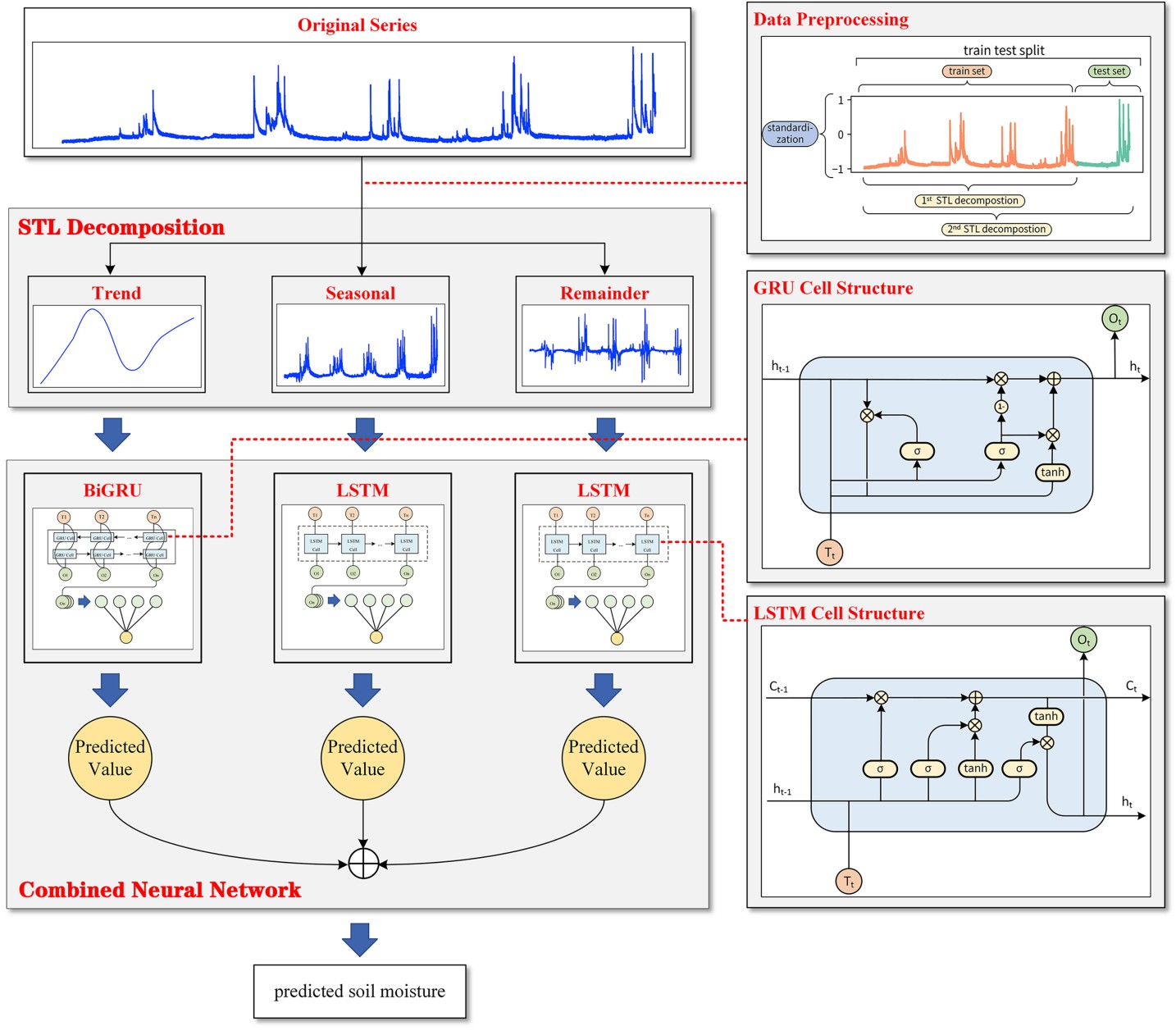

**Figure 4 The overall framework of the model.**

decomposition, and the original sequence is decomposed into the trend component, seasonal component and remainder component. During decomposition, to avoid data leakage and prove the effectiveness of the model, the subsequence as a training set was first decomposed alone, and then the whole sequence was decomposed to obtain the test set. Then, the BiGRU model is used for the obtained trend component, and an LSTM model is used to fit the timing information for the seasonal component and the remainder

component. Finally, the combined model extracts the hidden layer state of the last time step of each cyclic neural network model and outputs the predicted values of the three components through a fully connected layer. STL decomposed the sequence in an additive way, which made it convenient to model the three components independently.

The predicted values of the three components were added to obtain the final prediction results for the plateau soil moisture content.

## EXPERIMENTAL ANALYSIS

### Performance metrics

In this experiment, the root mean square error (RMSE), mean absolute error (MAE) and adjusted goodness of fit (adjusted $R^2$) were used to compare the experimental results output by each model and judge the model's single-step and short-range prediction performance. Smaller values of RMSE and MAE indicate higher model accuracy. The closer $R^2$ is to 1, the higher the prediction accuracy of the model is, and the adjusted $R^2$ eliminates the influence of sequence length and the number of features in the model on the index so that the $R^2$ of different models can be compared with each other. The calculation formulas of RMSE, MAE, and adjusted $R^2$ are shown in Eqs. (10)–(13), respectively.

$$RMSE = \sqrt{\frac{1}{m}\sum_{i=1}^{m}\left(y_{test}^{(i)} - \hat{y}_{test}^{(i)}\right)^2} = \sqrt{MSE} \quad \#() \tag{12}$$

$$MAE = \frac{1}{m}\sum_{i=1}^{m}\left|y_{test}^{(i)} - \hat{y}_{test}^{(i)}\right| \quad \#() \tag{13}$$

$$R^2 = 1 - \frac{\sum_i \left(\hat{y}^{(i)} - y^{(i)}\right)^2}{\sum_i \left(\bar{y} - y^{(i)}\right)^2} \quad \#() \tag{14}$$

$$adjusted\ R^2 = 1 - \frac{(1 - R^2)(n - 1)}{n - p - 1} \quad \#() \tag{15}$$

where $y^{(i)}$, $\hat{y}^{(i)}$ and $\bar{y}$ represent the true value, the model estimated value and the sample sequence mean, respectively. $n$ is the sequence length, and $p$ is the number of features in the model.

In long-term forecasting (*e.g.*, when the prediction horizon is 24 h), it is crucial to assess the accuracy of the predicted sequences in both the temporal and shape aspects, in addition to evaluating the "point-to-point" accuracy using the above three indicators. This study employs the dynamic time warping (DTW) metric based on the Euclidean distance, as proposed by *Sakoe & Chiba (1978)*, to evaluate the accuracy in the shape aspect.

The temporal distortion index (TDI) introduced by *Frías-Paredes et al. (2017)* is utilized to measure the accuracy in the temporal aspect. Smaller values of DTW and TDI indicate a higher prediction accuracy of the model.
**Table 1  Main parameter settings of the BiGRU and LSTM models.**

| Parameter | Value |
| --- | --- |
| Predicted time window size | $24 \times 365$ |
| Batch size | 200 |
| Training rounds | 100 |
| Number of hidden layer neurons | 32 |
| Number of model layers | 1 |
| Loss function | MSE |
| Activation function | ReLU |
| Optimizer | Adam |

## Experimental environment and parameter setting

The experimental environment adopted in this article is an Intel Xeon 8358P 2.6 GHz CPU and NVIDIA RTX A5000 GPU, and the model is built based on PyTorch under Python 3.8.

The early stop mechanism is introduced in the first pretraining. When the training model loss function is without gain in 10 iterations, the iteration will be stopped. This measure can not only ensure the fitting accuracy of the model but also effectively prevent overfitting and save the training time of the model. The results of pretraining show that the model generally achieves the optimal effect when the number of iterations is approximately 80. Therefore, the training cycle is set as 100 in the subsequent experiment in this article. The results of pretraining also show that due to the powerful fitting ability of BiGRU and LSTM models, the model with a simple structure can already achieve sufficient fitting ability under the problem studied in this article, while the overly complex model structure will make the performance worse. To make the model obtain as much historical information as possible and exclude too much noise at the same time, the prediction window size was set as 1 year, that is, $24 \times 365$ h. Based on various considerations, the main super parameters and training parameters set in the model training process are shown in Table 1.

## Experimental results and analysis

### STL decomposition results

The plateau soil moisture data used in this study have an obvious annual cycle, and the data sampling frequency is once per hour. Therefore, the cycle length parameter $n_p$ is set as $24 \times 365$ h, and the parameter $n_s$ is set to 7, which is slightly larger than the number of cycles contained in the data. The parameter $n_t$ is determined according to the empirical rule described in Section 2.2.1. The three components obtained by STL decomposition are shown in Fig. 5.

According to the decomposition results, the STL algorithm can adequately extract the trend and periodic information contained in the sequence, and the seasonal term clearly shows the periodic variation in soil moisture in the plateau. The remainder sequence has a mean value of 0 and fluctuates randomly nearby, which also proves that the STL

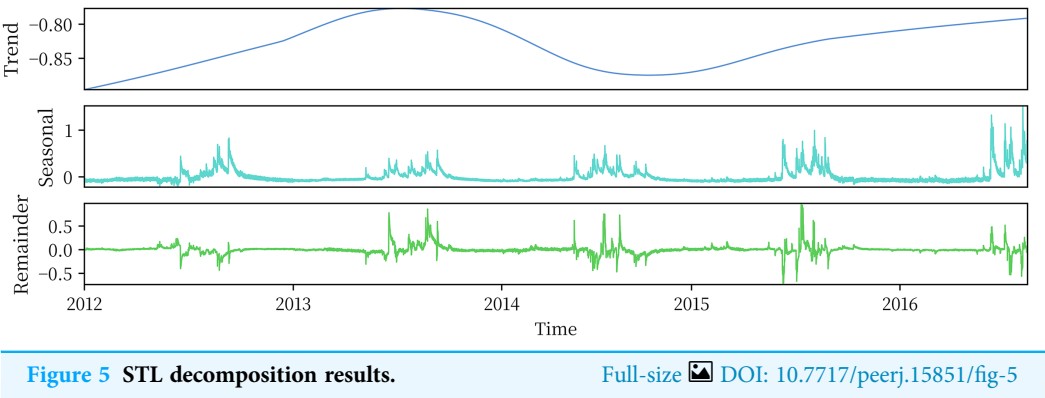

**Figure 5  STL decomposition results.**     

decomposition adopted is effective. Figure 5 also shows that the plateau soil moisture showed an increasing trend during 2012–2016, but there was a low trough during 2014–2015.

### Prediction performance of the combined model

Figure 6 shows the effect of the depth model on the test set for prediction of the three components obtained by STL decomposition. For Figs. 6A–6C, when the data points are scattered as closely as possible along the diagonal representing perfectly accurate predictions, it indicates higher prediction accuracy of the model. From the figure, it can be observed that the model achieves a good fit and prediction accuracy for the soil moisture content data in the plateau region. With few exceptions, most data points fall along the diagonal. Figures 6D–6E are used to observe the distribution of residuals obtained for each component. The variance of the prediction error of the model is extremely small for all three components and the mean value is extremely close to 0. This indicates that the three components obtained by STL decomposition can be effectively handled by the deep recurrent neural network structure.

The comparative experimental data in Table 2 and Fig. 7 show that for the trend component, the BiGRU model used in this article is the best, while for the seasonal component and the remainder component, the adopted LSTM model has the best performance.

The STL-BiGRU, STL-LSTM and STL-RNN are chosen to compare and validate the use of the combined models for the overall soil moisture content series on the QTP. The LSTM, CNN-BiGRU and LSTM-Attention models, which are commonly used in time series and multivariate soil moisture prediction models, are also selected as comparative models due to the lack of 'end-to-end' prediction models for soil moisture prediction in existing studies. The performance of these three commonly used models in combination with STL decomposition is also examined. The prediction series and evaluation metrics obtained from each model are shown in Fig. 8 and Table 3, respectively.

The predicted values given by the reanalysis method ERA5 can roughly reflect the trend of soil moisture content, but there is a large gap compared to the measured values. The comparison models, although closer to the measured values, not only have larger errors in prediction, but also tend to significantly overestimate or underestimate the

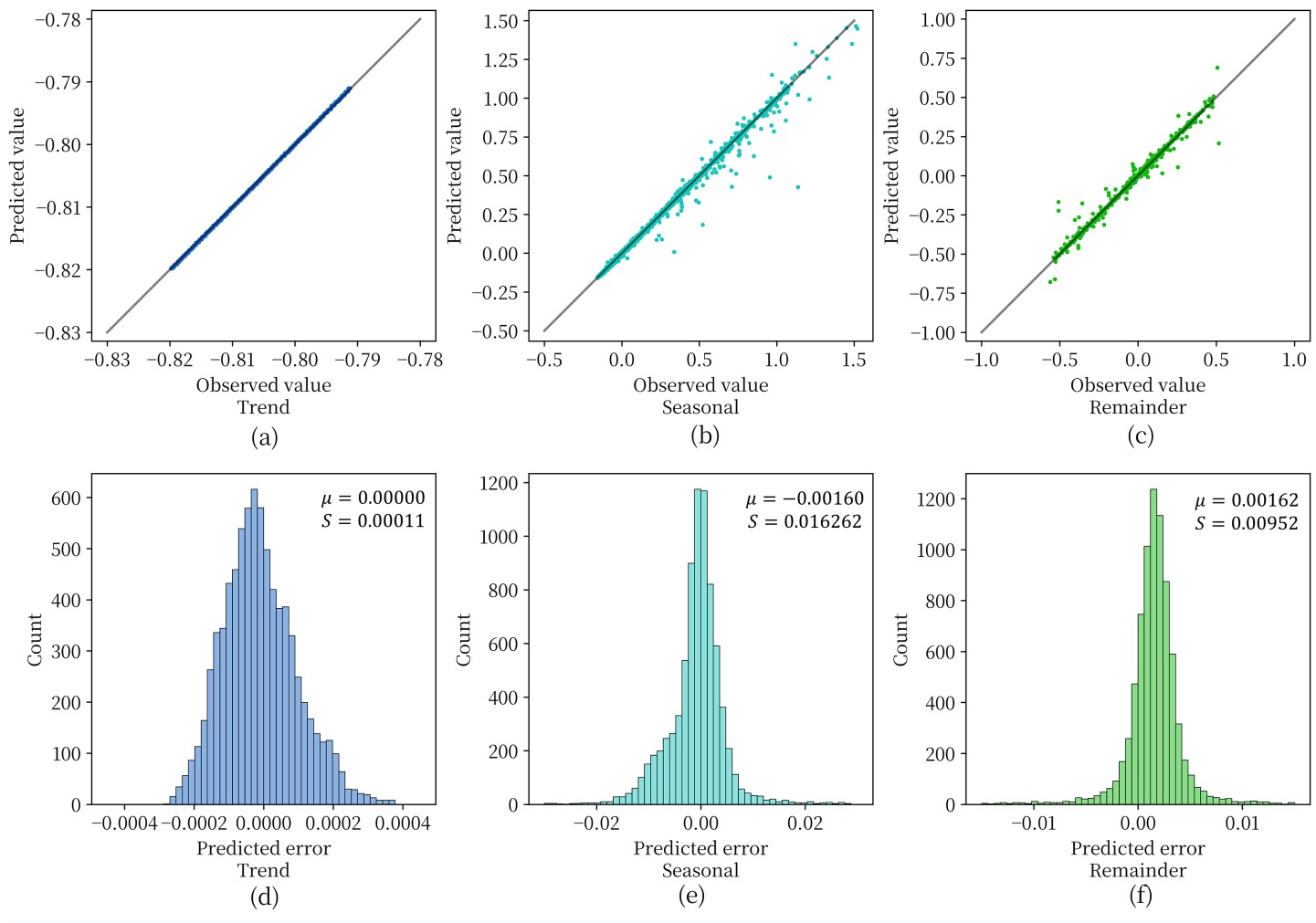

**Figure 6 The prediction error of the model for the three components.**

**Table 2 RMSE predicted by different models for each component.**

|  | Trend | Seasonal | Remainder |
|---|---|---|---|
| GRU | 0.00018 | 0.01669 | 0.00965 |
| BiGRU | 0.00011 | 0.01675 | 0.00962 |
| LSTM | 0.00013 | 0.01605 | 0.00949 |
| BiLSTM | 0.00015 | 0.01667 | 0.00958 |
| RNN | 0.00019 | 0.01814 | 0.00983 |
| CNN-BiGRU | 0.00401 | 0.04619 | 0.02450 |

sudden changes in soil moisture content. The proposed combined model in this article achieves the best results among all the compared models, and the RMSE metrics are reduced by 4.72–22.92% compared to the STL-RNN, STL-BiGRU, and STL-BiLSTM models using only a single depth model, which proves the effectiveness of the combined

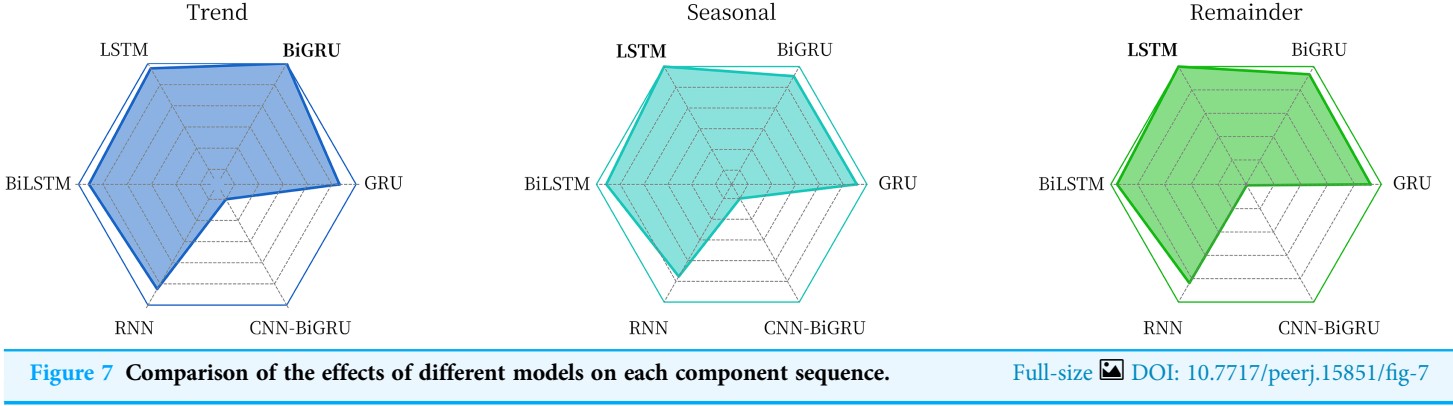

**Figure 7 Comparison of the effects of different models on each component sequence.**

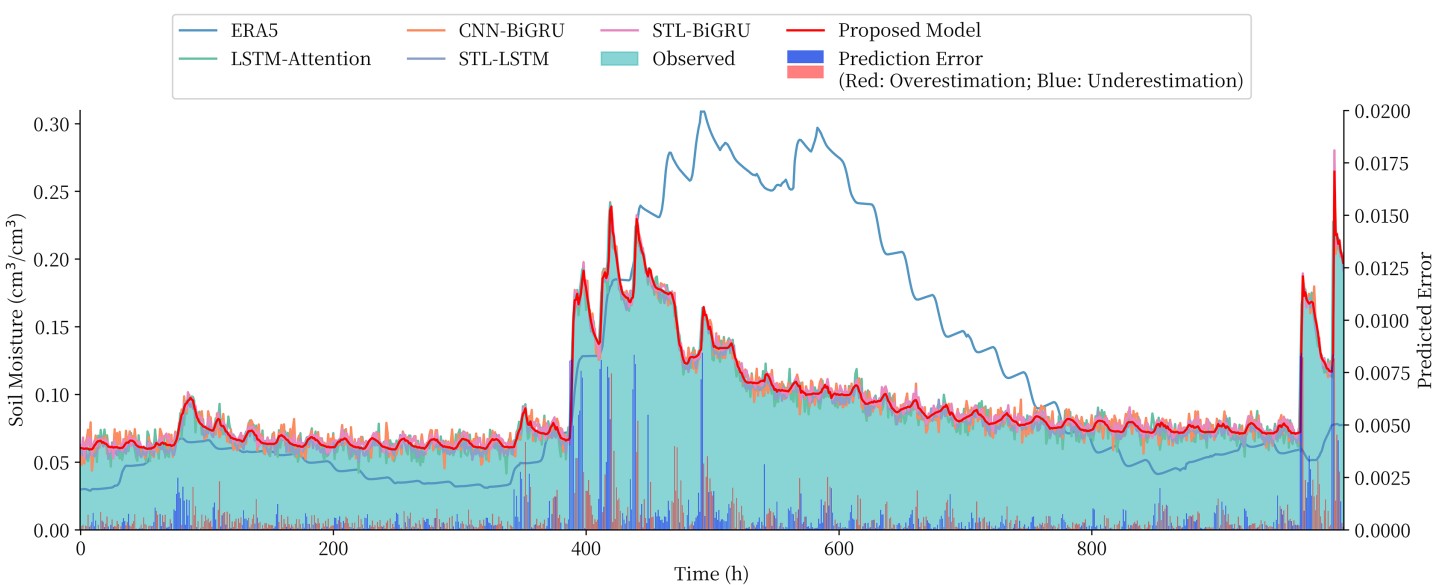

**Figure 8 Comparison of the prediction of different model and the prediction error of the proposed model.**

**Table 3 Comparison of evaluation metrics across models for single-step forecasting.**

| Model | RMSE | MAE | adjusted $R^2$ |
|---|---|---|---|
| STL-BiGRU-LSTM | 0.01936 | 0.00462 | 0.99330 |
| STL-RNN | 0.02032 | 0.00501 | 0.99160 |
| STL-BiGRU | 0.02069 | 0.00552 | 0.99220 |
| STL-LSTM | 0.02512 | 0.00679 | 0.99276 |
| STL-CNN-BiGRU | 0.05287 | 0.02138 | 0.94997 |
| STL-LSTM-attention | 0.02612 | 0.00830 | 0.98778 |

model approach. There was also a 7.72–28.27% performance improvement over the undecomposed LSTM, LSTM-Attention and CNN-BiGRU and a more significant improvement over STL-CNN-BiGRU and STL-LSTM-Attention with more complex structures.

**Table 4 RMSE of models with different prediction step sizes.**

| Model | 2 h | | 8 h | | 16 h | | 24 h | |
|---|---|---|---|---|---|---|---|---|
| | RMSE | Δ | RMSE | Δ | RMSE | Δ | RMSE | Δ |
| Proposed model | 0.02854 | – | 0.06105 | – | 0.08131 | – | 0.09426 | – |
| STL-LSTM | 0.03281 | +14.96% | 0.06967 | +14.12% | 0.09148 | +12.51% | 0.10393 | +10.26% |
| STL-BiGRU | 0.03031 | +6.20% | 0.06687 | +9.53% | 0.09194 | +13.07% | 0.1012 | +7.36% |
| LSTM | 0.03564 | +24.88% | 0.06925 | +13.43% | 0.10361 | +27.43% | 0.11277 | +19.64% |
| LSTM-Attention | 0.03525 | +23.51% | 0.06423 | +5.21% | 0.08627 | +6.10% | 0.10532 | +11.73% |
| CNN-BiGRU | 0.03102 | +8.69% | 0.06636 | +8.70% | 0.08787 | +8.07% | 0.10208 | +8.30% |

**Table 5 Comparison of DTW and TDI of each model.**

| | Proposed model | STL-LSTM | STL-BiGRU | LSTM | LSTM-attention | CNN-BiGRU |
|---|---|---|---|---|---|---|
| DTW | 0.12798 | 0.13164 | 0.13808 | 0.14144 | 0.25624 | 0.21491 |
| TDI | 0.47729 | 0.56860 | 0.74114 | 0.61623 | 0.98432 | 0.85565 |

In this article, we mainly use RMSE to evaluate the multistep prediction performance of the model, while DTW and TDI are used to examine the long-range prediction performance of the model (prediction horizon of 24 h), and the results are shown in Tables 4 and 5.

The proposed combined model achieves the best results in all prediction horizons. The RMSE of the combined model improved by 6–25% over the single models with STL processing for prediction steps of 2 h, 8 h, 16 h, and 24 h and by 5–10% over the models without STL processing. For DTW and TDI, the proposed model also achieves the best value at the most extreme prediction horizon of 24 h, demonstrating the performance of the proposed model for multistep prediction.

To further quantitatively assess the stability of the model in making predictions at different forecast horizons, drawing from the work of *Zhang et al. (2018)*, this study introduces the S index as shown in Eq. (14):

$$S = \frac{1}{D} \sum_{i=1}^{D} \frac{R_i - R_0}{d_i} \#() \tag{16}$$

where $D$ represents the number of different forecast horizons tested, $d_i$ denotes the $i$th forecast horizon, $R_i$ represents the RMSE of the model at the $i$th forecast horizon, and $R_0$ represents the RMSE of the model for a single-step forecast. The S index for each model is presented in Table 6.

## GENERALIZED PERFORMANCE ANALYSIS

Aiming to further investigate the generalization performance of the proposed model, in this part, the proposed combined model is used to make a fitting prediction of the soil heat

**Table 6 Comparison of prediction stability S of each model.**

|  | Proposed model | STL-LSTM | STL-BiGRU | LSTM | LSTM-attention | CNN-BiGRU |
|---|---|---|---|---|---|---|
| $S \times 1{,}000$ | 4.20 | 4.21 | 4.60 | 4.49 | 5.03 | 4.41 |

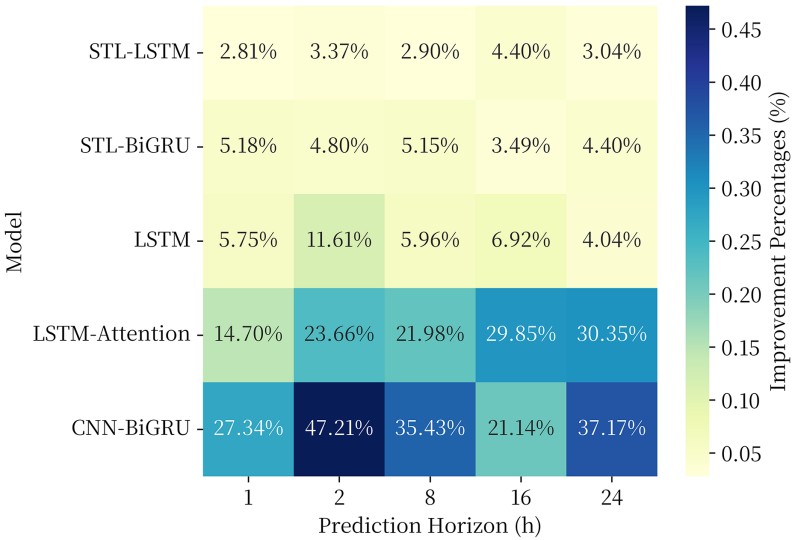

**Figure 9 RMSE of each model in the generalization performance experiment.**

flux time series data, and the same comparison model is selected as in Section 3.3. The soil heat flux time series data were obtained from the National Qinghai-Tibet Plateau Scientific Data Center (http://dx.doi.org/10.11888/Meteoro.tpdc.270910). In this article, observations from the BJ site of the Naqu Station of Plateau Climate and Environment (NPCE-BJ) at a soil depth of 10 cm during 2007–2013 were taken (*Zhu et al., 2019*). The data were not missing in the selected time period. The study of *Ma et al. (2020)* shows that this series also has obvious periodic and trend changes, and its data characteristics are similar to the soil moisture content series investigated in this article, which is suitable to be used as the dataset for generalization performance analysis. The experimental results are shown in Fig. 9.

The results of the generalization performance experiments show that the models also exhibit optimal results over the comparison models on the new data set. The combined model proposed in this article generally shows a 3–5% performance improvement over the single model in the comparison model at 1, 2, 8, 16, and 24 h prediction horizon, and a 10–50% performance improvement over the LSTM, LSTM-Attention, and CNN-BiGRU models commonly used in time series. The experimental results provethat the proposed model has strong generalization ability in the field of geography and climate of QTP.

## CONCLUSION

In this article, a combined prediction model based on STL decomposition and a deep recurrent neural network is proposed to address the complex characteristics of soil

moisture content time series on the Tibetan Plateau. The proposed model achieves "end-to-end" prediction through a simple and clear structure, thus requiring no additional complex feature engineering or other information input. This article introduces STL decomposition to the field of soil moisture prediction for the first time and demonstrates that the decomposition can effectively extract and separate long-term trend variation, periodic seasonal variation and random perturbation of soil moisture series in the plateau. The three component series obtained from the STL decomposition are extracted and fitted by a BiGRU and two LSTM models, and the best results are obtained. The RMSE of the combined model proposed in this article reaches 0.01936, and the adjusted $R^2$ reaches 0.99330, which is a 5–30% performance improvement over the single model or existing models. Meanwhile, the model proposed in this article demonstrates the best stability in different prediction steps, especially in making long-range predictions, and the model proposed in this article can balance the accuracy in predicting sequence morphology and time domain. It is demonstrated that the STL-based neural network combination model has high accuracy, robustness and effectiveness for soil moisture sequences in the plateau, which has high practical application value and shows the feasibility of applying deep learning methods to soil moisture prediction in the plateau. The proposed method also has reference value for other complex natural system time series prediction problems, such as soil state indicator sequences.

### Funding
The research is supported by the Open Project Program of State Key Laboratory of Crop Biology (2021KF04). The publication costs of the article were funded by the Shandong Province Natural Science Foundation (ZR2019PC046). The funders had no role in study design, data collection and analysis, or decision to publish.

### Grant Disclosures
The following grant information was disclosed by the authors:
State Key Laboratory of Crop Biology: 2021KF04.

### Competing Interests
The authors declare that they have no competing interests.

### Author Contributions

- Lufei Zhao conceived and designed the experiments, performed the experiments, analyzed the data, prepared figures and/or tables, authored or reviewed drafts of the article, and approved the final draft.
- Tonglin Luo conceived and designed the experiments, performed the experiments, analyzed the data, prepared figures and/or tables, authored or reviewed drafts of the article, and approved the final draft.

- Xuchu Jiang conceived and designed the experiments, performed the experiments, analyzed the data, prepared figures and/or tables, authored or reviewed drafts of the article, and approved the final draft.
- Biao Zhang conceived and designed the experiments, performed the experiments, analyzed the data, prepared figures and/or tables, authored or reviewed drafts of the article, and approved the final draft.

## Data Availability

Code and raw data are available at GitHub and Zenodo: https://github.com/camel712/Prediction-of-soil-moisture-using-BiGRU-LSTM-model-with-STL-decomposition.

Haoyu Zhang. (2023). qwaeqa/Prediction-of-PM2.5-concentration-based-on-the-CEEMDAN-RLMD-BILSTM-LEC-model: v1.0.0 (v1.0.0). Zenodo. https://doi.org/10.5281/zenodo.8140823.

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
