# Peer review of "Prediction of soil moisture using BiGRU-LSTM model with STL decomposition in Qinghai–Tibet Plateau"

_PeerJ, doi:10.7717/peerj.15851_

## Round 0.1 · original submission · Major Revisions

Reviewers have suggested improvements to the paper that are critical. Please follow each comment carefully in revising your manuscript. Some specific points raised include
- testing validity of results
- establishing uniqueness of finding
- comparison to literature
- raw data availability
- discuss spatial component of data
- quality of figures
- writing errors

·

Basic reporting

The paper discusses the use of time series analysis and prediction methods and highlights the limitations of traditional mathematical modeling methods in capturing complex relationships and suggests the application of machine learning and deep learning techniques for time series analysis and prediction. Support vector machines (SVM) and recurrent neural networks (RNN) are mentioned as examples of such methods.

Some minor points:

1. Can the authors prove this "The Ali Network data based on the Tibetan Plateau can provide representative coverage of the climate and surface hydrometeorological conditions in the cold and arid region of the Qinghai-Tibet Plateau (QTP)".  The spatial component is missing in this paper.

2. The predicted results should be compared with satellite and reanalysis soil moisture data (SMAP, ERA5, etc), as  satellite data exists over the period.

3. The quality of the figures is not good. provide map parameters (North, lat and long , etc), for Fig1, for instance.

Experimental design

No comment

Validity of the findings

No comment

Reviewer 2 ·

Basic reporting

1. The language is below expectation and the authors should improve the manuscript quality with a fluent speaker or a professional English editing.
2. The authors fail to present a comprehensive literature review and identify the research challenges and gaps to fill in this study.
3. The raw data are not shared.

Experimental design

1. The technical contributions are weak. All the models are from well-known techniques.
2. If BiGRU and LSTM are considered, what about GRU and BiLSTM?
3. The authors should compare with previous studies and state-of-the-art solutions, besides the model variants.

Validity of the findings

1. The data are not provided.
2. The results cannot be replicated. The authors try to deceive the reviewers by providing only part of the code and processed data. This behavior should not be encouraged.

Reviewer 3 ·

Basic reporting

The reviewer could have provided specific examples or highlighted specific sections of the manuscript where the clarity and professional English usage are particularly strong. This would have added more depth to the evaluation and helped the authors identify the specific strengths of their writing.

Experimental design

no comment

Validity of the findings

The paper lacks an assessment of its impact and novelty within the field. While the research methodology and findings are presented, the authors did not explicitly discuss the potential implications and significance of their work. It is important for the authors to clearly state the rationale and benefit of their research to the existing literature, highlighting any novel aspects of their approach.

---

## Round 0.2 · Minor Revisions

Please add the following files for the replication of results.

from myutil.dataset import FenLiangSteps
from myutil.model_0 import GTrend,LTrend
from myutil.train import train,test

Reviewer 2 ·

Basic reporting

no comment

Experimental design

The results cannot be replicated. The authors are still trying to deceive the reviewers by providing only part of the code, once again in the revised version.

Important files are missing as pointed out as follows:
from myutil.dataset import FenLiangSteps
from myutil.model_0 import GTrend,LTrend
from myutil.train import train,test

The results are not reliable and the authors are trying to hide their code.

Validity of the findings

no comment

---

## Round 0.3 · accepted · Accept

Thanks for the revision and updating the GitHub site for scripts.